The genome and transcriptome of Phalaenopsis yield insights into floral organ development and flowering regulation

Huang Jian-Zhi 1
Lin Chih-Peng 2 4
Cheng Ting-Chi 1
Huang Ya-Wen 1
Tsai Yi-Jung 1
http://orcid.org/0000-0002-1028-3839 Cheng Shu-Yun 1
Chen Yi-Wen 1
Lee Chueh-Pai 2
Chung Wan-Chia 2
Chang Bill Chia-Han 2 3 bchang@yourgene.com.tw
Chin Shih-Wen 1 swchin@mail.npust.edu.tw
http://orcid.org/0000-0003-2807-176X Lee Chen-Yu 1 culee@mail.npust.edu.tw
Chen Fure-Chyi 1 fchen@mail.npust.edu.tw
1 Department of Plant Industry, National Pingtung University of Science and Technology , Pingtung , Taiwan
2 Yourgene Bioscience , Shu-Lin District, New Taipei City , Taiwan
3 Faculty of Veterinary Science, The University of Melbourne , Parkville, Victoria , Australia
4 Department of Biotechnology, School of Health Technology, Ming Chuan University , Gui Shan District, Taoyuan , Taiwan
McCormick Sheila
Electronic publication date: 2016 May 12
Publication date: 2016
Volume: 4
Electronic Location ID: e2017
Received 2016 Feb 2; Accepted 2016 Apr 17
Copyright: © 2016 Huang et al.
Copyright year: 2016
Copyright holder: Huang et al.
License: This is an open access article distributed under the terms of the Creative Commons Attribution License, which permits unrestricted use, distribution, reproduction and adaptation in any medium and for any purpose provided that it is properly attributed. For attribution, the original author(s), title, publication source (PeerJ) and either DOI or URL of the article must be cited.
License URL: https://creativecommons.org/licenses/by/4.0/

Keywords: Phalaenopsis, Draft genome, PhAGL6b, Flower organ development, Flowering time

Funding: Agriculture and Food Agency, Council of Agriculture, Taiwan 102AS-9.1.1-FD-Z2(1), 103AS-9.1.1-FD-Z2(1), and 104AS-9.1.1-FD-Z2(1) This work was supported by grants from the Agriculture and Food Agency, Council of Agriculture, Taiwan (grant numbers 102AS-9.1.1-FD-Z2(1), 103AS-9.1.1-FD-Z2(1), and 104AS-9.1.1-FD-Z2(1)). The funders had no role in study design, data collection and analysis, decision to publish, or preparation of the manuscript.

==============================
The Phalaenopsis orchid is an important potted flower of high economic value around the world. We report the 3.1 Gb draft genome assembly of an important winter flowering Phalaenopsis ‘KHM190’ cultivar. We generated 89.5 Gb RNA-seq and 113 million sRNA-seq reads to use these data to identify 41,153 protein-coding genes and 188 miRNA families. We also generated a draft genome for Phalaenopsis pulcherrima ‘B8802,’ a summer flowering species, via resequencing. Comparison of genome data between the two Phalaenopsis cultivars allowed the identification of 691,532 single-nucleotide polymorphisms. In this study, we reveal that the key role of PhAGL6b in the regulation of labellum organ development involves alternative splicing in the big lip mutant. Petal or sepal overexpressing PhAGL6b leads to the conversion into a lip-like structure. We also discovered that the gibberellin pathway that regulates the expression of flowering time genes during the reproductive phase change is induced by cool temperature. Our work thus depicted a valuable resource for the flowering control, flower architecture development, and breeding of the Phalaenopsis orchids.

Introduction

Phalaenopsis is a genus within the family Orchidaceae and comprises approximately 66 species distributed throughout tropical Asia (Christenson, 2002). The predicted Phalaenopsis genome size is approximately 1.5 gigabases (Gb), which is distributed across 19 chromosomes (Lin et al., 2001). Phalaenopsis flowers have a zygomorphic floral structure, including three sepals (in the first floral whorl), two petals and the third petal develops into a labellum in early stage of development, which is a distinctive feature of a highly modified floral part in second floral whorl unique to orchids. The gynostemium contains the male and female reproductive organs in the center (Rudall & Bateman, 2002). In the ABCDE model, B-class genes play important role to perianth development in orchid species (Chang et al., 2010; Mondragón-Palomino & Theissen, 2011; Tsai et al., 2004). In addition, PhAGL6a and PhAGL6b, expressed specifically in the Phalaenopsis labellum, were implied to play as a positive regulator of labellum formation (Huang et al., 2015; Su et al., 2013). However, the relationship between the function of genes involved in floral-organ development and morphological features remains poorly understood.

Phalaenopsis orchids are produced in large quantity annually and are traded as the most important potted plants worldwide. During greenhouse production of young plants, the high temperature > 28 °C was routinely used to promote vegetative growth and inhibit spike initiation (Blanchard & Runkle, 2006). Conversely, a lower ambient temperature (24/18 °C day/night) is used to induce spiking (Chen et al., 2008) to produce flowering plants. Spike induction in Phalaenopsis orchid by this cool temperature is the key to precisely controlling its flowering date. Several studies have indicated that cool temperature during the night are necessary for Phalaenopsis orchids to flower (Blanchard & Runkle, 2006; Chen et al., 1994; Chen et al., 2008; Wang, 1995). Despite a number of expressed sequence tags (ESTs), RNA-seqs and sRNA-seqs from several tissues of Phalaenopsis have been reported and deposited in GenBank or OrchidBase (An & Chan, 2012; An, Hsiao & Chan, 2011; Hsiao et al., 2011; Su et al., 2011), only a few flowering related genes or miRNAs have been identified and characterized. In addition, the clues to the spike initiation during reproductive phase change in the shorten stem, which may produce signals related to flowering during cool temperature induction, have not been dealt with. At this juncture, the molecular mechanisms leading to spiking of Phalaenopsis has yet to be elucidated.

Here we report a high-quality genome and transcriptomes (mRNAs and small RNAs) of Phalaenopsis Brother Spring Dancer ‘KHM190,’ a winter flowering hybrid with spike formation in response to cool temperature. We also provide resequencing data for summer flowering species P. pulcherrima ‘B8802.’ Our comprehensive genomic and transcriptome analyses provide valuable insights into the molecular mechanisms of important biological processes such as floral organ development and flowering time regulation.

Methods Summary

The genome of the Phalaenopsis Brother Spring Dancer ‘KHM190’ cultivar was sequenced on the Illumina HiSeq 2000 platform. The obtained data were used to assemble a draft genome sequence using the Velvet software (Zerbino & Birney, 2008). RNA-Seq and sRNA-Seq data were generated on the same platform for genome annotation and transcriptome and small RNA analyses. Repetitive elements were identified by combining information on sequence similarity at the nucleotide and protein levels and by using de novo approaches. Gene models were predicted by combining publically available Phalaenopsis RNA-Seq data and RNA-Seq data generated in this project. RNA-Seq data were mapped to the repeat masked genome with Tophat (Trapnell, Pachter & Salzberg, 2009) and CuffLinks (Trapnell et al., 2012). The detailed methodology and associated references are available in Appendix S1.

Results and Discussion

Genome sequencing and assembly

We sequenced the genome of the Phalaenopsis orchid cultivar ‘KHM190’ (Appendix S1, Fig. S1a) using the Illumina HiSeq 2000 platform and assembled the genome with the Velvet assembler, using 300.5 Gb (90-fold coverage) of filtered high-quality sequence data (Appendix S1, Table S1). This cultivar has an estimated genome size of 3.45 Gb on the basis of a 17 m depth distribution analysis of the sequenced reads (Appendix S1, Figs. S2 and S3; Tables S2 and S3). De novo assembly of the Illumina reads resulted in a sequence of 3.1 Gb, representing 89.9% of the Phalaenopsis orchid genome. Following gap closure, the assembly consisted of 149,151 scaffolds (≥ 1,000 bp), with N50 lengths of 100 and 1.5 kb for the contigs. Approximately 90% of the total sequence was covered by 6,804 scaffolds of > 100 kb, with the largest scaffold spanning 1.4 Mb (Appendix S1, Tables S3–S5 and Data S17). The sequencing depth of 92.5% of the assembly was more than 20 reads (Appendix S1, Fig. S3), ensuring high accuracy at the nucleotide level. The GC content distribution in the Phalaenopsis genome was comparable with that in the genomes of Arabidopsis (The Arabidopsis Genome Initiative, 2000), Oryza (International Rice Genome Sequencing Project, 2005 and Vitis (Jaillon et al., 2007) (Appendix S1, Fig. S4).

Gene prediction and annotation

Approximately 59.74% of the Phalaenopsis genome assembly was identified as repetitive elements, including long terminal repeat retrotransposons (33.44%), DNA transposons (2.91%) and unclassified repeats (21.99%) (Appendix S1, Fig. S5 and Table S6). To facilitate gene annotation, we identified 41,153 high-confidence and medium-confidence protein-coding regions with complete gene structures in the Phalaenopsis genome using RNA-Seq (114.1 Gb for a 157.6 Mb transcriptome assembly), based on 15 libraries representing four tissues (young floral organs, leaves, shortened stems and protocorm-like bodies (PLBs)) (Appendix S1, Table S7 and Data S18), and we used transcript assemblies of these regions in combination with publically available expressed sequence tags (Su et al., 2011; Tsai et al., 2013) for gene model prediction and validation (Data S1–S2). We predicted 41,153 genes with an average mRNA length of 1,014 bp and a mean number of 3.83 exons per gene (Table 1 and Data S3). In addition to protein coding genes, we identified a total of 562 ribosomal RNAs, 655 transfer RNAs, 290 small nucleolar RNAs and 263 small nuclear RNAs in the Phalaenopsis genome (Appendix S1, Table S8). We also obtained 92,811,417 small RNA (sRNA) reads (18–27 bp), representing 6,976,375 unique sRNA tags (Appendix S1, Fig. S6 and Data S6–S7). A total of 650 miRNAs distributed in 188 families were identified (Data S8), and a total of 1,644 miRNA-targeted genes were predicted through the alignment of conserved miRNAs to our gene models (Appendix S1, Fig. S7 and Data S9–S10).

Table 1 Statistics of the Phalaenopsis draft genome.

Estimate of genome size	3.45 Gb	
Chromosome number (2n)	38	
Total size of assembled contigs	3.1 Gb	
Number of contigs (≥ 1 kbp)	630,316	
Largest contig	50,944	
N50 length (contig)	1,489	
Number of scaffolds (≥ 1 kbp)	149,151	
Total size of assembled scaffolds	3,104,268,398	
N50 length (scaffolds)	100,943	
Longest scaffold	1,402,447	
GC content	30.7	
Number of gene models	41,153	
Mean coding sequence length	1,014 bp	
Mean exon length/number	264 bp/3.83	
Mean intron length/number	3,099 bp/2.83	
Exon GC (%)	41.9	
Intron GC (%)	16.1	
Number of predicted miRNA genes	650	
Total size of transposable elements	1,598,926,178	

The Phalaenopsis gene families were compared with those of Arabidopsis (The Arabidopsis Genome Initiative, 2000), Oryza (International Rice Genome Sequencing Project, 2005), and Vitis (Jaillon et al., 2007) using OrthoMCL (Li, Stoeckert & Roos, 2003). We identified 41,153 Phalaenopsis genes in 15,855 families, with 8,532 gene families being shared with Arabidopsis, Oryza and Vitis. Another 5,143 families, containing 12,520 genes, were unique to Phalaenopsis (Fig. 1). In comparison with the 29,431 protein-coding genes estimated for the Phalaenopsis equestris genome (Cai et al., 2015), our gene set for Phalaenopsis ‘KHM190’ contained 11,722 more members, suggesting a more wider representation of genes in this work. This difference in gene number may be due to different approaches between Phalaenopsis ‘KHM190’ and Phalaenopsis equestris. Besides, Phalaenopsis ‘KHM190’ is a hybrid while P. equestris species, which may show gene number difference due to different genetic background. To better annotate the Phalaenopsis genome for protein-coding genes, we generated RNA-seq reads obtained from four tissues as well as publically available expressed sequence tags for cross reference. We defined the function of members of these families using (The Gene Ontology Consortium, 2008), the Kyoto Encyclopedia of Genes and Genomes (KEGG) (Kanehisa et al., 2012) and Pfam protein motifs (Finn et al., 2014) (Fig. 2; Data S3–S5 and S19).

Figure 1 Venn diagram showing unique and shared gene families between and among Phalaenopsis, Oryza, Arabidopsis and Vitis.

Figure 2 GO (A) and Pfam (B) annotation of Phalaenopsis protein-coding genes.

The genes in the High confidence (HC) and Medium Confidence (MC) gene sets were functionally annotated based on homology to annotated genes from the NCBI non-redundant database (Data S3). The functional domains of Phalaenopsis genes were identified by comparing their sequences against protein databases, including (The Gene Ontology Consortium, 2008), KEGG (Kanehisa et al., 2012) and Pfam (Finn et al., 2014; Finn, Clements & Eddy, 2011) databases. GO terms were obtained using the Blast2GO program (Conesa & Gotz, 2008). In the GO annotations, 16,034, 27,294 and 16,360 genes were assigned to the biological process, cellular component, and molecular function categories, respectively (Fig. 2A). Based on KEGG pathway mapping, we were able to assign a significant proportion of the Phalaenopsis gene sets to KEGG functional or biological pathway categories (11,452 sequences; 140 KEGG orthologous terms) (Data S4). To investigate protein families, we compared the Pfam domains of Phalaenopsis genome. A total of 1,842 Pfam domains were detected among the Phalaenopsis sequences. The most abundant protein domains in Phalaenopsis genome were pentatricopeptide repeats (PPRs, pfam01535), followed by the WD40 (pfam00400), EF hand (pfam00036) and ERM (Ezrin/radixin/moesin, pfam00769) domains (Fig. 2B and Data S5). Furthermore, conserved domains could be identified in 50.17% of the predicted protein sequences based on comparison against Pfam databases. In addition, we identified 2,610 transcription factors (TFs) (6.34% of the total genes) and transcriptional regulators in 55 gene families (Appendix S1, Figs. S8–S10 and Datas S11–S12).

Regulation of Phalaenopsis floral organ development

The relative expression of all Phalaenopsis genes was compared through RNA-Seq analysis of shoot tip tissues from shortened stems, leaf, floral organs and PLB samples, in addition to vegetative tissues, reproductive tissues, and germinating seeds from P. aphrodite (Su et al., 2011; Tsai et al., 2013) (Appendix S1, Fig. S12 and Data S1). Phalaenopsis orchids exhibit a unique flower morphology involving outer tepals, lateral inner tepals and a particularly conspicuous labellum (lip) (Rudall & Bateman, 2002). However, our understanding of the regulation of the floral organ development of the genus is still in its infancy. To comprehensively characterize the genes involved in the development of Phalaenopsis floral organs, we obtained RNA-Seq data for the sepals, petals and labellum of both the wild-type and peloric mutant of Phalaenopsis ‘KHM190’ at the 0.2 cm floral bud stage, at which shows early sign of labellum differentiation. This mutant presented an early peloric fate in its lateral inner tepals. In a peloric flower, the lateral inner tepals are converted into a lip-like morphology at this young bud stage (Appendix S1, Figs. S11B and S12A). We identified 3,743 genes that were differentially expressed in the floral organs of the wild-type and peloric mutant plants. Gene Ontology analysis of the differentially expressed genes in Phalaenopsis floral organs revealed functions related to biological regulation, developmental processes and nucleotide binding, which were significantly altered in both genotypes (Huang et al., 2015). TFs seem to play a role in floral organ development. Of the 3,309 putative TF genes identified in the Phalaenopsis genome showed differences in expression between the wild-type and peloric mutant plants (Data S11).

MADS-box genes are of ancient origin and are found in plants, yeasts and animals (Trobner et al., 1992). This gene family can be divided into two main lineages, referred to as types I and II. Type I genes only share sequence similarity with type II genes in the MADS domain (Alvarez-Buylla et al., 2000). Most of the well-studied plant genes are type II genes and contain three domains that are not present in type I genes: an intervening (I) domain, a keratin-like coiled-coil (K) domain, and a C-terminal (C) domain (Munster et al., 1997). These genes are best known for their roles in the specification of floral organ development, the regulation of flowering time and other aspects of reproductive development (Dornelas et al., 2011). In addition, MADS-box genes are also widely expressed in vegetative tissues (Messenguy & Dubois, 2003; Parenicova et al., 2003). The ABCDE model comprises five major classes of homeotic selector genes: A, B, C, D and E, most of which are MADS-box genes (Theissen, 2001). However, research on the ABCDE model was mainly focused on herbaceous plants and has not fully explained how diverse angiosperms evolved. The function of many other genes expressed during floral development remains obscure. Phalaenopsis exhibits unique flower morphology involving three types of perianth organs: outer tepals, lateral inner tepals, and a labellum (Rudall & Bateman, 2002). Despite its unique floral morphological features, the molecular mechanism of floral development in Phalaenopsis orchid remains largely unclear, and further research is needed to identify genes involved in floral differentiation. Recently, several remarkable research studies on Phalaenopsis MADS-box genes have revealed important roles of some of these genes in floral development, such as four B-class DEF-like MADS-box genes that are differentially expressed between wild-type plants and peloric mutants with lip-like petals (Tsai et al., 2004) and a PI-like gene, PeMADS6, that is ubiquitously expressed in petaloid sepals, petals, columns and ovaries (Tsai et al., 2005).

In the Phalaenopsis genome sequence assembly, a total of 122 genes were predicted to encode MADS-box family proteins (Appendix S1, Fig. S8 and Data S12). To obtain a more accurate classification, phylogenetic trees were constructed via the neighbour-joining method, with 1000 bootstraps using MEGA5 (Tamura et al., 2011). The differentially expressed genes (DEGs) among 122 Phalaenopsis MADS-box genes were obtained from our Phalaenopsis RNA-Seq data (Data S11). The expression profile indicated that most MADS-box genes are widely expressed in diverse tissues. These results will be helpful in elucidating the regulatory roles of these genes in the Phalaenopsis floral organ development.

Notably, we previously reported one of the MADS-box genes, PhAGL6b, upregulated in the peloric lateral inner tepals (lip-like petals) and lip organs (Huang et al., 2015). To understand the expression mode, we therefore cloned the full-length sequence of PhAGL6b from lip organ cDNA libraries for the wild-type, peloric mutant and big lip mutant. The big lip mutant developed a petaloid labellum instead of the regular lip observed in the wild-type flower (Fig. 3B). Interestingly, we identified four alternatively spliced forms of PhAGL6b that were specifically expressed only in the petaloid labellum of the big lip mutant (Figs. 3C and 3D; Appendix S1 and Fig. S11). To determine whether the alternatively spliced forms of PhAGL6b affect the conversion of the labellum to a petal-like organ in the big lip mutant, we performed RT-PCR of total RNA extracted from the labellum organs of plants with different big lip mutant phenotypes and wild-type plants (Appendix S1, Table S11 and Fig. 4A) to amplify the PhAGL6b transcripts. Interestingly, among all of the big lip mutant phenotypes, 500–700 bp bands were detected, corresponding to PhAGL6b alternatively spliced forms, which were not found in any of the other orchid plants (Fig. 4A). We further examined the expression of PhAGL6b and its alternatively spliced forms in the labellum organs of Phalaenopsis plants with different big lip phenotypes and wild-type plants via real-time PCR (Appendix S1, Table S11). In the big lip mutants, the expression of native PhAGL6b was reduced by 42–70%, whereas all of the alternatively spliced forms were expressed more strongly compared with the wild-type plants (Fig. 4B). In summary, the RT-PCR and real-time PCR experiments corroborated the specific expression of the alternatively spliced forms of PhAGL6b in the petal-like lip of big lip mutants. Thus, PhAGL6b might play crucial role in the development of the labellum in Phalaenopsis.

Figure 3 Possible evolutionary relationship of PhAGL6b in the regulation of lip formation and floral symmetry in Phalaenopsis orchid.

(A) Wild-type flower. (B) A big lip mutant of Phalaenopsis World Class ‘Big Foot.’ (C) Representative RT-PCR result showing the mRNA splicing pattern of PhAGL6b in wild-type (W) and big lip mutant (M). (D) Alignment of the amino acid sequences of alternatively spliced forms of PhAGL6b. (E) Model of PhAGL6b spatial expression for controlling Phalaenopsis floral symmetry. Ectopic expression of PhAGL6b in the distal domain (petal; pink), petal converts into a lip-like structure that leads to radial symmetry. Ectopic expression in proximal domain, (sepal; blue) sepal converts into a lip-like structure that leads to bilateral symmetry. The alternative processing of PhAGL6b transcripts produced in proximal domain (labellum; pink), labellum converts into a petal-like structure that leads to radial symmetry. PhAGL6b expression patterns in Phalaenopsis floral organs are either an expansion or a reduction across labellum. This implies that PhAGL6b may be a key regulator to the bilateral or radially symmetrical evolvements. Pink color: 2nd whorl of the flower; blue color: 1st whorl of the flower.

Figure 4 Different labellum types of wild-type and big lip mutant Phalaenopsis flowers.

RT-PCR analysis of the mRNA splicing pattern of PhAGL6b in wild-type plants (98201-WT1 and 98201-WT2) and different big lip mutant types (A). Splicing variants of PhAGL6b, as detected via qRT-PCR in the labellum organ of different big lip mutant types (B).

The four isoforms of the encoded PhAGL6b products differ only in the length of their C-terminus region (Fig. 3D). C-domain is important for the activation of transcription of target genes (Honma & Goto, 2001) and may affect the nature of the interactions with other MADS-box proteins in multimeric complexes (Geuten et al., 2006; Gramzow & Theissen, 2010). In Oncidium, L (lip) complex (OAP3-2/OAGL6-2/OAGL6-2/OPI) is required for lip formation (Hsu et al., 2015). The Phalaenopsis PhAGL6b is an orthologue of OAGL6-2. In our study, the PhAGL6b and its different spliced forms may each other compete the Phalaenopsis L-like complex to affect labellum development as reported in Oncidium (Hsu et al., 2015). This provides a novel clue further supporting the notion that PhAGL6b may function as a key floral organ regulator in Phalaenopsis orchids, with broad impacts on petal, sepal and labellum development (Fig. 3E).

Control of flowering time in Phalaenopsis

The flowering of Phalaenopsis orchids is a response to cues related to seasonal changes in light (Wang, 1995), temperature (Blanchard & Runkle, 2006) and other external influences (Chen et al., 1994). A cool night temperature of 18–20 °C for approximately four weeks will generally induce spiking in most Phalaenopsis hybrids, while high temperature inhibits it. To compare gene expression between a constant high-temperature (30/27 °C; day/night) and inducing cool temperature (22/18 °C), we collected shoot tip tissues from shortened stems of mature P. aphrodite plants after treatment at a constant high temperature (BH) and a cool temperature (BL) (1–4 weeks) for RNA-Seq data analysis (Appendix S1, Figs. S12G–S12I). More than 7,500 Phalaenopsis genes were found to be highly expressed in the floral meristems during the 4 successive cool temperature periods (showing at least a 2-fold difference in the expression level in the BL condition relative to BH) (Data S13). The identified flowering-related genes correspond to transcription factors and genes involved in signal transduction, development and metabolism (Fig. 3 and Data S14). The classification of these genes includes the following categories: photoperiod, gibberellins (GAs), ambient temperature, light-quality pathways, autonomous pathways and floral pathway integrators (Fornara, de Montaigu & Coupland, 2010; Mouradov, Cremer & Coupland, 2002). However, the genes involved in the photoperiod, ambient temperature, light quality and autonomous pathways did not show significant changes in the floral meristems during the cool temperature treatments (Appendix S1, Fig. S13 and Data S14). By contrast, the expression patterns of genes involved in pathways that regulate flowering, comprising a total of 22 GA pathway-related genes, were related to biosynthesis, signal transduction and responsiveness. The GA pathway-related genes and the floral pathway integrator genes have been revealed as representative key players in the link between flowering promotion pathways and the floral transition regulation network in several plant species (Mutasa-Göttgens & Hedden, 2009). In contrast to the expression patterns observed in BL and BH, the GA biosynthetic pathway and positively acting regulator genes showed high expression levels in BL. Furthermore, the expression level of negatively acting regulators, like DELLA genes identified, was suppressed by the cool temperature which allowing the activation of flowering related genes. The genes included in the flowering promotion pathways and floral pathway integrators were generally upregulated in BL (Figs. 5 and 6; Data S11). These findings suggest that the GA pathway may play a crucial role in the regulation of flowering time in Phalaenopsis orchid during cool temperature.

Figure 5 Expression profiles of genes of flowering time regulation pathway with high temperature and cool temperature treatment.

Only the genes with twofold change in expression during cool temperature treatments are revealed.

Figure 6 Predicted pathway in the regulation of spike induction in Phalaenopsis.

Red indicates that the involved genes are more highly expressed in the GA biosynthesis pathway; pink gene names indicate their differential expression in the GA response pathway. Blue gene names represent the activation of flower architecture genes. Red arrows show the steps of the GA signaling stage; Pink arrows direct the steps of inflorescence evocation stage; Blue arrows reveal the steps of flower stalk initiation stage. Inverted T indicates the genes downregulated 2X over. GA20ox, GA3ox, GAMYB, FT, SOC1, LFY and AP1 are upregulated 2X over.

Genetic polymorphisms for Phalaenopsis orchids

The Phalaenopsis genome assembly also provides the basis for the development of molecular marker-assisted breeding. Analysis of the Phalaenopsis genome revealed a total of 532,285 simple sequence repeats (SSRs) (Appendix S1, Fig. S14, Table S9 and Data S15). To enable the identification of single nucleotide polymorphisms (SNPs), we re-sequenced the genome of a summer flowering species, P. pulcherrima ‘B8802,’ with about tenfolds coverage. Comparison of the genome data from the two Phalaenopsis accessions (KHM190 and B8802) allowed the discovery of 691,532 SNPs, which should be valuable for future development of SNP markers for Phalaenopsis marker-assisted selection (Appendix S1, Fig. S15, Table S10 and Data S16). P. pulcherrima is an important parent for small flower and summer-flowering cultivars in breeding program. These SNP markers may contribute valuable tools for varietal identification, genetic linkage map development, genetic diversity analysis, and marker-assisted selection breeding in Phalaenopsis orchid.

Conclusion

In this study, we sequenced, de novo assembled, and extensively annotated the genome of one of the most important Phalaenopsis hybrids. We also annotated the genome with a wealth of RNA-seq and sRNA-seq from different tissues, and many genes and miRNAs related to floral organ development, flowering time and protocorm (embryo) development were identified. Importantly, this RNA-Seq and sRNA-seq data allowed us to further improve the genome annotation quality. In addition, mining of SSR and SNP molecular markers from the genome and transcriptomes is currently being adopted in advanced breeding programs and comparative genetic studies, which should contribute to efficient Phalaenopsis cultivar development. Despite that the P. equestris genome has been reported recently (Cai et al., 2015), focus on floral organ development and flowering time regulation has not been dealt with. In our study, we obtained transcriptomes from shortened stems (which initiate spikes in response to low ambient temperature) and floral organs, and generated valuable data on potentially regulating flowering time key genes and floral organ development. The genome and transcriptome information of our work should provide a constructive reference resource to upgrade the efficiency of cultivation and the genetic improvement of Phalaenopsis orchids.

Supplemental Information

Supplemental Information 1 Dataset_1-14.

Click here for additional data file.

Supplemental Information 2 Dataset_S13.

Click here for additional data file.

Supplemental Information 3 Dataset_S15.

Click here for additional data file.

Supplemental Information 4 Dataset_S16-1.

Click here for additional data file.

Supplemental Information 5 Dataset_S16-2.

Click here for additional data file.

Supplemental Information 6 Dataset_S17.

Click here for additional data file.

Supplemental Information 7 Dataset S18.

Sequence read archive.

Click here for additional data file.

Supplemental Information 8 Dataset_S19.

Click here for additional data file.

Supplemental Information 9 Supplementary Information Appendix.

Click here for additional data file.

Additional Information and Declarations

Competing Interests

Author Contributions

DNA Deposition

Data Deposition

The authors declare that they have no competing interests.

Chih-Peng Lin, Chueh-Pai Lee, Wan-Chia Chung and Bill Chia-Han Chang are employees of Yourgene Bioscience, Taiwan.

Jian-Zhi Huang conceived and designed the experiments, performed the experiments, analyzed the data, contributed reagents/materials/analysis tools, wrote the paper, prepared figures and/or tables, reviewed drafts of the paper.

Chih-Peng Lin performed the experiments, analyzed the data, contributed reagents/materials/analysis tools, prepared figures and/or tables, reviewed drafts of the paper.

Ting-Chi Cheng performed the experiments.

Ya-Wen Huang performed the experiments.

Yi-Jung Tsai performed the experiments.

Shu-Yun Cheng performed the experiments.

Yi-Wen Chen performed the experiments.

Chueh-Pai Lee performed the experiments, contributed reagents/materials/analysis tools.

Wan-Chia Chung performed the experiments, contributed reagents/materials/analysis tools.

Bill Chia-Han Chang analyzed the data, contributed reagents/materials/analysis tools.

Shih-Wen Chin conceived and designed the experiments, analyzed the data, contributed reagents/materials/analysis tools, wrote the paper, prepared figures and/or tables, reviewed drafts of the paper.

Chen-Yu Lee conceived and designed the experiments, analyzed the data, contributed reagents/materials/analysis tools, wrote the paper, prepared figures and/or tables, reviewed drafts of the paper.

Fure-Chyi Chen conceived and designed the experiments, analyzed the data, contributed reagents/materials/analysis tools, wrote the paper, prepared figures and/or tables, reviewed drafts of the paper.

The following information was supplied regarding the deposition of DNA sequences:

SRR1747138, SRR1753943, SRR1753944, SRR1753945, SRR1753946, SRR1753947, SRR1753948, SRR1753949, SRR1753950, SRR1752971, SRR1753106, SRR1753165, SRR1753166 SRR1762751, SRR1762752, SRR1762753, SRR1760428, SRR1760429, SRR1760430, SRR1760432, SRR1760433, SRR1760435, SRR1760436, SRR1760438, SRR1760439, SRX396172, SRX396784, SRX396785, SRX396786, SRX396787, SRX396788 SRR1760091, SRR1760211, SRR1760212, SRR1760213, SRR1760270, SRR1760271, SRR1760523, SRR1760524, SRR1760525, SRR1760526, SRR1760527, SRR1760528, SRR1760530, SRR1760531, SRR1760532.

The following information was supplied regarding data availability:

The research in this article did not generate any raw data.

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
