# Peer review of "The genome and transcriptome of Phalaenopsis yield insights into floral organ development and flowering regulation"

_PeerJ, doi:10.7717/peerj.2017_

## Round 0.1 · original submission · Major Revisions

In principle your paper appears to be a valuable addition to the field, but needs substantial revision to make the manuscript more user-friendly and useful to the research community. Please address all the comments of the reviewers - I think their suggestions will greatly improve your manuscript.

Reviewer 1 ·

Basic reporting

I have some general comments about the manuscript as a whole, but then comment specifically on its component parts in the ‘General Comments to the Author’ section. This paper presents an impressive body of work, but still needs substantial revision before it will be ready for publication. The English is pretty good, but this manuscript would benefit from some further editing for grammar. In general, more detail/discussion needs to be added throughout the manuscript. I would suggest that a number of figures should be moved from supplemental data to the main manuscript, and the existing figures need more primary data included and/or further clarifying detail/explanation.
This manuscript hasn’t been inappropriately subdivided, but hasn’t been written as a cohesive whole. It will need some re-writing/re-focusing to pull together its various elements.
Although the appropriate data was available for review, it wasn’t clear to me whether the sequence/SSR data had been deposited to any public short read archives.

Experimental design

This paper reports some original primary research, particularly the new Phalaenopsis genome, some interesting data on alternative splicing in big lip mutants, and some interesting transcriptomics on floral induction. However, the floral organ transcriptomics seem to have been reported before in Huang, 2015 (PLoS ONE).
The research question is not clearly defined, particularly not in the abstract. This is in part because the paper consists of four disparate parts that have not been integrated into a cohesive whole – I discuss this more below.
Although the methods in the supplementary information seem complete and technically sound, the methods summary in the main text is inadequate. More details are necessary in the main manuscript.

Validity of the findings

Overall, the data presented in this paper seems robust and statistically sound. Apart form the reported floral transcriptomics (see more discussion below), this paper presents novel results, and does not represent a simple repetition of well-known results. The authors’ conclusions are solidly grounded in their results and the authors don’t overstate their conclusions. However, as I mention above, it is not clear where the data has been deposited (which public databases have been utilized?).
This manuscript would actually benefit from a little more speculation – more discussion of the authors’ results, and some speculation on their data’s biological significance would be welcomed.

Additional comments

General Comments for the Author
Although the authors have produced a number of impressive datasets, this paper was a bit of a challenge to review because it consists of four component parts that are all fairly disparate, with distinct motivations. These four component parts have all been incompletely developed, and have not been integrated into a cohesive whole. I will discuss each of these sections in turn below. As I see it, the four parts are:
1. The genome of Phalaenopsis KHM190 cultivar
2. Floral organ transcriptomics comparing wild type and peloric mutants
3. Transcriptomics of plants under inducing and non-inducing temperatures
4. SSR development for breeding


1. The Genome
The genome section is sorely underdeveloped. A single Venn diagram of shared gene families between Phaleonopsis and 4 other plants is all that is shown in the single ‘genome’ figure (Fig. 1). Although it will be challenging to mold this paper into a more cohesive whole, I think the whole paper could be anchored by more in-depth discussion of the new genome they have sequenced, with some side-trips into floral morphology, the induction of flowering, and marker-assisted breeding. To this end, I would suggest moving some of the supplemental data into the main body of the paper, to be discussed at more length. For example, Fig. S7 and S8 could move into the main manuscript. The gene phylogenies, and the discussion of the gene families investigated, could also be moved into the main body of the text, but they would need to be more carefully constructed – ideally using either maximum likelihood or Bayesian phylogenetic analysis. Although the authors say they ran bootstrap replicates for the MADS box genes, no support values are shown on any of the trees.

In addition, it was very difficult to assess the quality of the new genome without more comparisons to the already-published Phalaenopsis equestris genome (Cai, 2015). At the very least I would have liked to see P. equestris included in the Fig. 1 Venn diagram. The authors state that they recovered more protein coding genes in comparison to P. equestris either because of a different experimental approach or because KHM190 is a hybrid, but then leave it at that. Which genes and gene families are over-represented in the hybrid genome? Are there any interesting patterns? I don’t know how genetically divergent two species of Phalaenopsis might be, maybe it would not be unexpected to see wide divergence between the sequenced genomes, but none of this is discussed.

2. Floral Organ Transcriptomics
As far as I can tell, the floral organ transcriptomics have already been reported in Huang 2015 (PLoS ONE). What this paper adds is more extensive characterization of one candidate gene – PhAGL6b – that is differentially spliced in big lip peloria. Since it has already been published elsewhere, I would suggest taking out all the floral organ transcriptome stuff, and spend more time discussing the big lip mutants and alternative splicing. I would suggest moving Fig. S14 and S15 into the main body of the manuscript. It is hard to tell from the paper – but I think the NPU-1458 and 98201 big lip mutants shown in S14 might be independently derived. If this is the case, this makes the alternative splicing story much more interesting. AGL6b had already been implicated in the specification of labellum identity (Hsu, 2015), but the results shown in Fig. S14 and S15 provide some evidence that alternative splicing of PhAGL6b may have been important in two separate derivations of a big lip mutant. This was not made clear in the results or the discussion. I would have appreciated more discussion of the big lip mutants. Are they genetically stable? If so, how do they segregate? If I am interpreting Fig. S14 correctly, the 98201 phenotype shows incomplete penetrance, what about the NPU-1458 phenotype? Is there anything in the sequence of AGL9 to suggest why this alternative splicing is happening?

In terms of the interpretation of the authors’ results in this section, I disagree on two points. (1) When it comes to describing these mutant phenotypes, I would argue that the big lip mutants are still zygomorphic - but labellum identity is partially lost. I realize that labellum morphology is a key component of overall floral zygomorphy, but I would say that AGL6b might be most correctly described as a labellum identity gene candidate, rather than as a floral zygomorphy gene candidate. (2) In the abstract, the authors state, ‘In this study, we reveal the key role of PhAGL6b in the regulation of flower organ development involves alternative splicing.’ I would argue that you don’t know whether alternative splicing is important in wild type floral development or not – but that there is correlative evidence that it may play a role in big lip mutants.

It took me quite a while to puzzle out Fig. 2e. I think it would be easier to interpret if the floral diagrams in the lower panel were labeled as ‘wild type’, ‘peloric’ and ‘big lip’. If possible, AGL6b expression indicated with shading and/or arrows rather than words could also help make interpretation easier. I think the authors use proximal and distal (rather than adaxial and abaxial) because of resupination. A clarifying line in the figure legend would be helpful.

3. Transcriptomics of floral induction
This section is represents the most novel part of the paper. Nonetheless, as with both of the previous sections, I would have appreciated more primary data in the main manuscript, not relegated to the supplementary information (e.g. Fig S16 could move into the main body of the manuscript). Fig. 3 is not easy to understand. It would definitely benefit from more labeling as well as from the inclusion of actual data on the figure, or in an earlier figure. In particular, it wasn’t clear to me what I was supposed to be seeing in the pictures of shoots in the upper panels. Gene expression patterns are very difficult to discern in Fig. 3. For example, Fig. S16 shows that DELLA expression is high under non-inducing temperatures, and then plummets in cooler temperatures. In Fig. 3, it could be seen as staying high under all conditions.

4. SSR marker development
This section is also underdeveloped. There was no mention of marker assisted breeding in Phalaenopsis before the SSRs are reported in the results section, there is no discussion of why P. pulcherrima B8802 was chosen for resequencing, and there was very little discussion of the SSR markers recovered. Is there even coverage of markers throughout the genome? Is marker-assisted breeding new in Phalaenopsis, or is this the first set of SSR markers that has been generated? Although I recognize that this set of SSRs could be a valuable resource for orchid breeders, I would have appreciated a little more discussion for a more general audience in both the introduction and the results sections.

Reviewer 2 ·

Basic reporting

I apologize to the authors for returning this review later than the designated deadline – and also to PeerJ for not meeting their usual standard.

Here is my review:

This is a very interesting paper describing analysis of a new genome assembly for a hybrid orchid.

I tried to look up the genome assembly sequence using the project identifier reported in the manuscript - PRJNA271641. It looks like only the plastid genome is available. Probably this just means the authors have not yet released the data to the public but will do so after the article is published. I really look forward to this new plant data set being available to the community.

In addition to the assembly, a core contribution of the paper is a set of structural (gene model) and functional annotations (GO terms, etc). In light of this, the authors should provide a BED file containing the gene models so they can easily be incorporated into public databases and used by the community for future RNA-Seq and related analyses. (A technical detail: non-protein coding gene models can report the same value for thickStart and thickEnd.)

For the convenience of readers, I hope the authors will also provide a fasta file containing the assembled genome sequence. Probably this will be provided as part of the genome sequence data that will be released with PRJNA271641.

I was able to look up the RNA-Seq accessions - http://trace.ncbi.nlm.nih.gov/Traces/sra/?study=SRP051837. Those data appear to be publicly accessible now.

The differential expression analysis was very interesting. However, I wished that the manuscript went into greater detail about how the experiments were designed. For example, how many replicates did you collect and what did your study consider a replicate? The Supplemental Methods talk about using DEGSeq (a Bioconductor package) to detect differential expression, but I wondered how realistic that was since it was not clear how many replicates were available.

Suggestion: Create a table that maps Short Read Archive sample and run identifiers onto sample types and replicate identifiers and include this in the manuscript. Get several people on your team to proof read it - carefully. Future members of the lab will appreciate being able to find this information in one place, as will readers who would like to repeat your analysis using newer software versions. If your data are important (which they are) readers will without a doubt want to do that as the tools for analyzing data are constantly improving and becoming easier to use.

I failed to do this myself in a similar paper and am now wishing I had been more diligent about data set curation as I could have saved my group and our collaborators much work.

Another suggestion: There were several places in the article where you referred to supplemental figures and files to make a key point. I think more text from the supplemental materials should be included in the main text. Readers will appreciate it if you move more of the supplementary material into the manuscript – it will be much easier to read.

Experimental design

See above.

Validity of the findings

See ablove.

Additional comments

See above.

---

## Round 0.2 · accepted · Accept

Thanks for making the revisions, the Manuscript is now accepted.